# TransPhos: A Deep-Learning Model for General Phosphorylation Site Prediction Based on Transformer-Encoder Architecture

**DOI:** 10.3390/ijms23084263

**Published:** 2022-04-12

**Authors:** Xun Wang, Zhiyuan Zhang, Chaogang Zhang, Xiangyu Meng, Xin Shi, Peng Qu

**Affiliations:** 1College of Computer Science and Technology, China University of Petroleum, Qingdao 266555, China; flyeagle237@163.com (Z.Z.); s20070030@s.upc.edu.cn (C.Z.); xiangyumeng@s.upc.edu.cn (X.M.); shix1104@163.com (X.S.); quupeng@163.com (P.Q.); 2State Key Laboratory of Computer Architecture, Institute of Computing Technology, Chinese Academy of Sciences, Beijing 100080, China

**Keywords:** phosphorylation site prediction, transformer, post-translational modifications

## Abstract

Protein phosphorylation is one of the most critical post-translational modifications of proteins in eukaryotes, which is essential for a variety of biological processes. Plenty of attempts have been made to improve the performance of computational predictors for phosphorylation site prediction. However, most of them are based on extra domain knowledge or feature selection. In this article, we present a novel deep learning-based predictor, named TransPhos, which is constructed using a transformer encoder and densely connected convolutional neural network blocks, for predicting phosphorylation sites. Data experiments are conducted on the datasets of PPA (version 3.0) and Phospho. ELM. The experimental results show that our TransPhos performs better than several deep learning models, including Convolutional Neural Networks (CNN), Long-term and short-term memory networks (LSTM), Recurrent neural networks (RNN) and Fully connected neural networks (FCNN), and some state-of-the-art deep learning-based prediction tools, including GPS2.1, NetPhos, PPRED, Musite, PhosphoSVM, SKIPHOS, and DeepPhos. Our model achieves a good performance on the training datasets of Serine (S), Threonine (T), and Tyrosine (Y), with AUC values of 0.8579, 0.8335, and 0.6953 using 10-fold cross-validation tests, respectively, and demonstrates that the presented TransPhos tool considerably outperforms competing predictors in general protein phosphorylation site prediction.

## 1. Introduction

Post-translational modifications (PTMs) are biochemical processes of proteins that take place post-translationally and are key mechanisms for regulating cellular function through covalent and general enzymatic modifications. PTMs are critical in regulating many biochemical reactions, such as protein synthesis, protein stability, and regulation of enzyme activity [1]. Protein phosphorylation is an important mechanism that regulates the activity of biological enzymes and is a very frequent type of PTMs [2]. Protein phosphorylation has important functions, especially in both prokaryotes and eukaryotes [3], which regulate many cellular processes, such as cell cycle regulation [4,5], protein–protein interaction [6], signal recognition [7], and DNA recovery [8]. More than a quarter of cellular proteins in eukaryotes are phosphorylated and modified, and more than half of them are responsible for various human diseases, especially near-reproductive diseases [9] and cancer [10]. It was found in recent research that protein phosphorylation is vital to understanding the signal regulation mechanism in cells and helping to develop new approaches to treat diseases caused by signal irregularity, such as cancer [11,12].

The prediction of phosphorylation sites is vital to the molecular mechanisms of biological processes associated with phosphorylation, which is of great help to disease-related research and drug design [13,14,15]. Experimental detection of protein phosphorylation sites is constantly advancing, with the earliest use of Edman degradation, followed by the development of mass spectrometry, which nowadays, in combination with Edman degradation, has become an effective tool for phosphoamine acid residue mapping in protein sequencing. Several traditional experimental methods have been adopted to identify phosphorylation sites, such as high-throughput mass spectrometry [16] and low-throughput 32P labeling [17,18].

Despite the unusually rapid development of proteomic technologies, comprehensive and exhaustive analysis of phosphorylated proteins remains difficult. Phosphorylation of proteins is an instable and dynamic process in the body, and there is a low abundance of phosphorylated proteins within the cell. The phosphate groups of phosphorylated proteins are easily lost during the isolation process and are difficult to protonate because of their electronegativity. Computational biology approaches have therefore become necessary and popular to handle the difficulties of experimental approaches for phosphorylation site prediction. 

Until now, more than 50 calculation methods for predicting phosphorylation sites have been proposed, a large number of which are based on machine learning approaches, such as Bayesian decision theory [19], support vector machines [20,21], random forests [22], and logistic regression [10]. For instance, Gao et al. [23] proposed a novel method called Musite by using local amino acid sequence frequencies, k-nearest neighbor features, and protein disorder scores to improve the prediction accuracy. Dou et al. [21] proposed an algorithm called PhosphoSVM, which combines several protein sequence properties with support vector machines to forecast phosphorylation sites.

These calculation methodologies and tools have facilitated the comprehension of phosphorylation and effectively improved performance. Most of them use multiple sequence-based features for multi-stage classification, such as physicochemical properties, protein disorder, and other areas of knowledge. In general, the use of extra tools may abstract redundancy features abstract, which is useful for the final prediction [22,24]. It needs to select some effective features. These selected features are applied to the machine learning algorithm for discriminative classification. So, end-to-end deep learning has made important breakthroughs in many fields, such as the transformer model in the field of machine translation [25]. The residual network effectively solves the problem of gradient disappearance in the training process of deep learning [26]. This makes it possible to train a deep learning classification model, which is used to predict protein phosphorylation sites. In a previous study, Luo et al. [27] proposed a tool named DeepPhos to predict phosphorylation sites.

In this study, a novel two-stage deep learning model, named TransPhos, is proposed to improve both the accuracy and Matthews correlation coefficient (MCC) of general protein phosphorylation prediction. In TransPhos, three encoders with the same structure and different window sizes based on the attention mechanism are designed. Instead of using any amino acid coding, we use the embedded layer to automatically learn an amino acid coding representation and then use multiple stacked encode layers to learn the vector representation of each amino acid. Each encode layer has the same structure as the encoder proposed by Vaswani et al. [25], but some parameters are modified. 

Two densely connected convolution neural network (DC-CNN) blocks that have the same window size are developed as the encoder. DC-CNN blocks with different window sizes and convolutional kernels can automatically learn the sequence features of protein phosphorylation sites. These features are concatenated into an intra-block connectivity layer (Inter-BCL) to further integrate the acquired information and finally provide predictions using the softmax function. To estimate the capabilities of TransPhos, we extracted many validated phosphorylation samples from two databases [28,29,30]. To verify the generalization of our model, the dataset Phospho. ELM was used as a training set and verification set, and the dataset from the PPA database was selected to test the performance. The experimental results demonstrated that TransPhos is superior to the existing general phosphorylation prediction methods in terms of AUC and MCC; compared with deep learning models, including CNN, LSTM, RNN, and FCNN; and some state-of-the-art deep learning-based prediction tools, including GPS2.1, NetPhos, PPRED, Musite, PhosphoSVM, SKIPHOS, and DeepPhos.

## 2. Results

TransPhos is a deep learning model that was developed to predict general phosphorylation sites. In this section, our model is compared with traditional deep learning models and other predictors. The results of the comparison with traditional deep learning models are described in Section 2.1, and the results of the comparison with other predictors are described in Section 2.2. It should be specified that the results on the training set were derived from 10-fold cross-validation. We performed significance F-tests on the prediction results of all models to demonstrate that our model predictions were significantly different from the other predictors, as described in Section 2.3.

### 2.1. Comparison with Different Deep Learning Models

We first compared TransPhos with several other deep learning models on the validation and test sets, including CNN, LSTM, RNN, and FCNN. The ROC curve is a very good tool to visualize the classification results, and the ROC curves on the S sites, when compared with the deep learning model on the training set, are shown in Figure 1. The ROC curves on the T sites and Y sites are shown in Figure A1 and Figure A2. Overall, our model achieved the highest Area Under Curve (AUC) values and exhibited a good performance.

Table 1 shows the details of the training set, where we used 10-fold cross-validation to select the optimal hyperparameters to avoid overfitting and to obtain enough feature information from the only available data. On the S sites, our model obtained the highest AUC value of 85.79%, which was 4.23, 1.79, 3.13, and 2.9% higher than CNN, LSTM, RNN, and FCNN, respectively. Besides the AUC values, we also calculated Accuracy (Acc), Sensitivity (Sn), Specificity (Sp), Precision (Pre), F1 Score (F1), and Matthews correlation coefficient (MCC) to measure the capabilities of our model. The calculation of these evaluation matrices is presented in Section 4.5. On the S sites, our model obtained the highest AUC values and the other metrics Acc, Sn, Pre, F1, and MCC were 78.18, 80.56, 76.83, 78.65%, and 0.564, respectively, which showed good performance. The Sp metric was only 1.36% lower than the best model FCNN. On the T sites, our model only showed the highest AUC and Sn metrics of 83.35 and 76.54%, respectively. The other metrics were slightly lower than the best model LSTM at this site. For the Y sites, our model showed the highest AUC value, F1 score, and MCC value. We used the PPA dataset as an independent test set to measure the performance of our model, and Table 2 shows the detailed results of the tests. The performance of our model was also very good on the T sites, with the highest AUC values and Acc and MCC, while the other metrics Sn, Sp, Pre, and F1 scores were 1.25, 3.21, 0.49, and 0.28% worse than the best results, respectively. 

Overall, our model performed best on the S sites and slightly worse on the T and Y sites, which may be due to the difficulty of training too many parameters in the encoder part and the poorer performance on smaller datasets. Other models also performed well on only one of the sites, so it can be assumed that our model performs better.

### 2.2. Comparison with Existing Phosphorylation Site Prediction Tools

Independent test datasets were collected from the PPA database in this study to measure the performance of the model. In this subsection, our model is compared with some other existing prediction tools, and the model parameters of all these predictors were obtained by 10-fold cross-validation on our training dataset P.ELM with their training strategies, facilitating a fair comparison. The left half of Table 3 shows the results of the 10-fold cross-validation, and the right half shows the results on the independent test set. We calculated the Sn, Sp, MCC, and AUC values to measure the model’s performance. Many well-known prediction tools were compared, including GPS2.1 [31], NetPhos [32], PPRED [33], Musite [23], PhosphoSVM [21], SKIPHOS [34], and DeepPhos [27]. The results showed that our model outperformed all other models for the S and T sites. For example, on the S sites, our model achieved the highest AUC values of 0.787 and 0.670 at GPS2.1, 0.643 at NetPhos, 0.676 at PPRED, 0.726 at Musite, 0.776 at PhosphoSVM, 0.691 at SKIPHOS, and 0.775 at DeepPhos. 

On the T sites, our model achieved the highest MCC value of 0.246 while the AUC value was only 0.002 lower than the optimal result. Our model did not perform the best on the Y sites, with SKIPHOS achieving the highest MCC and AUC values.

### 2.3. Significance Test of the Results

Regarding the results, most of the indicators of our model, such as ACC and MCC, performed better than other well-known predictors. However, many indicators were not as good as other predictor models. The significance F-test was used to demonstrate that our prediction results were significantly different from other forecasting models [35]. Usually, a *p*-value of less than 0.05 in the F-test indicates that the 2 statistical variables are significantly different [36]. As shown in Figure 2, we plotted the results of the statistical tests as a heat map, and the values in each box represent the corresponding *p*-values. The results of the significance test show that our model was significantly different from the predictions of most other models.

## 3. Discussion

In this work, we developed a deep learning model, named TransPhos, based on a transformer-encoder and CNN architecture, which can automatically learn features from protein sequences end to end to predict general phosphorylation sites. We performed 10-fold cross-validation on the training set and tested the model performance on an independent test set. Overall, our model performed extremely well on S and T sites, and our AUC values were the highest compared to other tools. Moreover, other major metrics were also significantly better than other models.

Firstly, we compared our model with several traditional deep learning models, including CNN, LSTM, FCNN, and RNN, on the test set. At the S sites, our model performed to the level system, and all evaluation metrics were the highest except Sp. AUC, Acc, Sn, Pre, F1, and MCC outperformed the other best models by 1.59%, 1.19%, 0.95%, 0.75%, 1.11%, and 0.23, respectively. A slight decrease in the performance of our model at the T sites was observed, but the main performance evaluation metrics, such as AUC, Acc, and MCC, were better than the other deep learning predictors: 0.6%, 0.56%, and 0.14% higher than the other best models, respectively. At the Y sites, our model’s performance was inferior to the other predictors.

Furthermore, we compared TransPhos with other current mainstream prediction models, including GPS2.1, NetPhos, PPRED, Musite, PhosphoSVM, SKIPHOS, and DeepPhos. Specifically, at the S sites, our model did not perform the best with other predictors, such as DeepPhos, as shown by the results of the 10-fold cross-validation. Our model achieved the best performance on the independent test set. The AUC and MCC values of our model on the test set were 0.8 and 0.7 percentage points higher than the other best models, respectively. This indicates that our model outperformed the comparison predictors in terms of the generalization performance. On the T sites, the AUC value of our model was only 0.2% lower than that of the best model DeepPhos, and the MCC value was 0.15 higher than that of DeepPhos, which indicates that our prediction results are much closer to the true value, judging from the results of the significance test. On the Y sites, neither our model nor the previous better performing model DeepPhos showed the best performance, and SKIPHOS obtained the highest MCC and AUC values at this site: 0.197 and 0.634, respectively.

Although our model showed a good performance in predicting the phosphorylation sites S and T, there are still some limitations that can be further improved. On the Y sites, since the total positive data of the Y site is much less compared to the S and T sites, and the encoder part of our model has an excess of parameters to be trained, this can easily cause model overfitting. To solve this problem, we used various approaches in the model design, such as regularization, the addition of dropout after the convolution layer [37,38], etc., but the limitations are still unresolved. Due to the excess of parameters and the limited access to kinase-specific phosphorylation site data, we conducted partial experiments and our model also performed poorly in kinase-specific phosphorylation site prediction. Thus, our model can only be used for general phosphorylation site prediction. In general, deep learning still performs poorly on small datasets [39,40]. However, in practical applications, there are far more S and T sites than Y sites, so the poor performance of the model on Y sites is acceptable [41].

From the results, the difference between our model and its predictors was not significant, so we performed a significance F-test to check the significance of our results with other predictor models [42]. Finally, we obtained the *p*-value of the test results. A *p*-value of less than 0.05 is usually considered as a significant difference between the 2 statistics. The results of our significance test are presented in Figure 2. From the significance test results, the following models were not significantly different from our model: CNN, FCNN, and GPS2.1 on the S sites; LSTM and GPS2.1 on the Y sites; and NetPhos and GPS2.1 on the T sites, respectively. A comparison of the prediction results showed that although several of the above models were statistically insignificant, our model showed a better prediction performance than these models at the corresponding loci. It can be concluded that the overall performance of our model was better than the existing models.

The main contribution of this study is the application of the encoder structure of the transformer to the phosphorylation prediction task [25]. Most previous studies have used either independent feature extraction followed by machine learning algorithms to predict phosphorylation sites [43] or one-hot encoding of protein sequences [27]. Feature extraction requires specialized domain knowledge and the use of one-hot encoding to effectively represent the interrelationships between protein sequences is difficult [44]. In this paper, the amino acid sequences of constituent proteins are first represented by dictionary encoding, then encoded into vector representation by the embedding layer, and then features are extracted by the encoder to further represent the effective information between sequences. After, convolutional neural networks are used to obtain the high-dimensional representation of phosphorylation sites, and finally classification is performed by the softmax function.

In summary, we present a deep learning architecture, TransPhos, that can be applied to general phosphorylation site prediction tasks to facilitate further biological research. The model has some uncertainties as the complete protein sequence is sliced into subsequences and predictions are then made for that subsequence. However, if a phosphorylation site is located at both ends of the whole protein sequence, then the sequence needs to be populated with a large number of identifiers, which can also lead to some unpredictable errors in the model when predicting such a site, such as prediction scores close to 0.5 and difficulty in distinguishing between positive and negative samples.

For future works, we will continue to work on phosphorylation site prediction, and we consider the use of an encoder-decoder architecture to train the whole protein sequence with the tag directly to achieve better prediction.

## 4. Materials and Methods

### 4.1. Overview

The overall architecture of TransPhos is described in Figure 3. We constructed our dataset, and the detailed process of data collection and preprocessing is described in Section 4.2. In Section 4.3, the structure and training process of our TransPhos model is described in detail and its performance on an independent test dataset is evaluated. In Section 4.4, we describe the training process of our model. Section 4.5 shows the performance evaluation used in this study.

### 4.2. Dataset Collection and Pre-Processing

#### 4.2.1. Dataset Collection

The construction of an effective benchmark dataset is crucial for the training and evaluation of deep learning models. PPA version 3.0 [28,29,45] and Phospho. ELM (P.ELM) version 9.0 [30] were used in this study. These two data sets were selected for two main reasons. The two datasets were utilized as benchmark datasets, which made comparison with other models easier. On the other hand, protein phosphorylation occurs in both animals and plants, the Phospho. ELM dataset includes phosphorylation sites from mammals, and the PPA dataset contains those from *Arabidopsis thaliana* (a plant). 

A total of 11,254 protein sequences were collected from the P.ELM dataset. Each sequence contains multiple protein phosphorylation sites, including 6635 serine (S) sites, 3227 threonine (T) sites, and 1392 tyrosine (Y) sites, respectively. The sites in the P.ELM database were extracted from other studies and phosphorylation proteomic analyses while the sites in the PPA database were experimentally measured by mass spectrometry. Some results predicted by computational methods are also available in the PPA database, and since some predictions have not been experimentally validated, only experimentally validated phosphorylation sites in PPA were used. In this study, BLASTClust [46] was used to cluster the protein in both datasets to remove redundant and duplicate protein sequences. We finally selected 12,810 proteins from the dataset to train the model.

#### 4.2.2. Data Pre-Processing

A complete protein sequence may comprise up to 4000 amino acids. In order to facilitate learning of the characteristics near the phosphorylation site, it is cut into subsequences with a window size of K, so that the amino acids in the middle of each subsequence are phosphorylation sites. If the length is insufficient, * is filled to ensure each subsequence has the same length. Other subsequences containing corresponding amino acids are also cut into subsequences with a length K. The middle of the sequence is the amino acids of non-phosphorylation sites. Such a setting will lead to an imbalance of positive and negative samples. We randomly deleted some negative samples to achieve the balance of positive and negative samples. Table 4 shows the number of sequences and phosphorylation sites that we used for this study.

### 4.3. Methods

TransPhos is a novel deep learning architecture that maps local protein sequences into high-dimensional vectors via a self-attentive mechanism, nonlinear transformations, and convolutional neural networks. The final classification result of phosphorylation sites is generated by the softmax function. TransPhos does not directly use a transformer encoder or a normal multilayer CNN but utilizes several encode layers with different window sizes and DC-CNN blocks. This allows for the efficient extraction of key protein sequence features for phosphorylation forecasting. 

For a protein represented by an amino acid sequence *x*, each amino acid $y\in D^{y}$, where y represents an amino acid and *D* is a dictionary encoding function that represents amino acids as digital. We sliced a sequence into sub-sequences of different window sizes and the position in the middle of the sequence is the phosphorylation site. For a protein sub-sequence x, the input of TransPhos with the total X Encoder is the set of vector $E^{x}\in R^{L_{x}\times I}$ for Encoder *x* (*x* = 1, 2, …, X), with $L_{x}$ and I being the corresponding local window size of phosphorylation sites and the size of the amino acid symbol vector, respectively. Here, I was set to 16. The input vector representation was obtained through an embedding layer by the dictionary code. In this study, we carefully studied various configurations of the model inputs with different window sizes and finally adopted a model configuration with a better performance with X=2 and window sizes of 31 and 51, which is slightly different from the predictors that had previously been proposed for phosphorylation sites [19,24,27,47] for Encoder 1 and 2, respectively. Therefore, the Encoder’s input shape was 33×16 and 51×16, respectively.

The Transphos model has two main stages. The first stage is X Encoders with several encoding layers. The encoder structure used in this paper was originally proposed by [25] in a machine translation task. In this study, the encoder parameters were fine-tuned to be applied to the phosphorylation prediction task.

Encoder: The encoder contains four structurally identical encode layers, each with two sub-layers. The first is a multi-head self-attention mechanism, and the second is a fully connected feed-forward network. The internal structure of the encoder is shown in Figure 4a.

The first sub-layer is an attention mechanism identical to the transformer’s encoder. The attentional function is described as:(1)$$\mathrm{Attention}\left(Q,K,V\right)=\mathrm{softmax}\left(\frac{QK^{T}}{\sqrt{d_{k}}}\right)\cdot V$$
where the matrices Quire (*Q*), Key (*K*), Value (*V*) are the inputs to the attention function, which contains a set of queries and keys of dimension $\sqrt{d_{k}}$, and values of dimension $d_{v}$. The output of the attention function is obtained by computing the dot product of the query with all keys, dividing each key by $\sqrt{d_{k}}$, and applying the softmax function and then multiplying it by values.

In practice, instead of using individual attention functions, we ran them in parallel, a design known as the multi-head attention mechanism [25], which is very helpful in improving the training speed. We calculated the output of the multi-head from the attention function as:(2)MultiHead(Q,K,V)=Concat(head1,…,headh)·WOwhere headi=Attention(QWiQ,KWiK,VWiV) 
where W is the parameter matrix WiQ∈ℝdmodel×dk,WiK∈ℝdmodel×dk,WiV∈ℝdmodel×dV and WO∈ℝhdv×dmodel.

In this task, we applied h = 4 parallel attention. For each layer, we set $d_{k}=d_{v}=d_{model}/h=4$. Since the number of all amino acid species was only 20, a shorter vector was used to represent them in this task. This design is advantageous to speed up the training, and to a certain extent to avoid rapid overfitting of the model on small data sets, which is especially important when training the Y site. Figure 5 illustrates the internal structure of the attention mechanism.

The encoder architecture of the transformer was used, hence the attention mechanism here is self-attentive, with the query, key, and value located in the same place. The input of the next encoder layer is sourced from the output of the previous encoder layer so that all the information of the previous encoder layer can be identified by the previous encoder layer. 

The first sub-layer is a fully connected feedforward network. It is defined as:(3)$$\mathrm{FFN}\left(x\right)=\max\left(0,xW_{1}+b_{1}\right)W_{2}+b_{2}$$

The output of the attention layer and the output of the feedforward neural network are connected with residual connections, and there is layer normalization [48] directly between the two sub-layers. 

After obtaining the output of the encoder, the second stage is X densely connected convolutional neural networks, the so-called DC-CNN blocks. We adopted several DC-CNN blocks with different window sizes and each DC-CNN block had the same structure. The internal construction of the DC-CNN block is shown in Figure 4b.

The input vector of the DC-CNN block is the output vector of the encoder, and the DC-CNN blocks perform a series of convolution operations to finally obtain a high-dimensional representation of the feature map. Each convolutional layer performs a one-dimensional convolutional operation along the length of the protein sequence, and after obtaining the corresponding output, an activation function is used to activate the neurons and implement the nonlinear transformation. Here, we used the ReLU activation function, which is very effective in convolutional neural networks. The feature maps obtained from the first convolutional layer are defined as:(4)$$h_{1}^{k}={\;a}_{k}\left(W^{k}E^{k}+b_{1}^{k}\right)$$
where $W^{k}$ represents the weight matrix with a size of $I\times S_{k}\times D$, I is the length of the vector representing individual amino acids in the protein sequence, and $S_{k}$ is the length of the convolution kernel. Here, *S* was set to 7, 13 and *k* was set to 1, 2. The number of convolutional layers is denoted by D, and we set it to 64. $b_{1}^{k}$ is the bias item. The dropout function was used after each convolution to randomly remove some neurons to reduce the risk of overfitting.

We adopted the Intra-BCLs to enforce the extraction of phosphorylated features in the DC-CNN block, connecting all previous convolutional layers with subsequent convolutional layers. Therefore, the output feature vectors of the ith convolutional layer in DC-CNN block k can be calculated as follows:(5)hik= ak(Wik[Ek,h1k,…,hi−1k]+bik) , i=2, 3
where Wik∈ℝD×Sk×D’ with D’ refers to the number of convolutional kernels in all convolutional layers in every DC-CNN block, and hi−1k represents the feature vectors generated by the (i − 1)th convolutional layer. 

After the sequence representation of the protein phosphorylation sites generated by the encoder and DC-CNN blocks is obtained, the next step uses the inter-BCL for concatenation along the first dimension as follows:(6)hf=[αk(hC1),αk(hC2)]
where hC1 and hC2 are the feature maps generated from the first and second DC-CNN blocks, respectively. Next, this feature map is transformed into a one-dimensional tensor by a flattened layer. A fully connected layer is connected, and the final prediction is performed by the softmax function:(7)P(y=1|x)=11+e−fcWc
(8)P(y=0|x)=1−11+e−fcWc
where Wc∈ℝfc×q, *q* refers to the number of categories to be predicted, which was set as 2. The predicted result is between 0 and 1.

### 4.4. Training of the TransPhos Model

Our model was trained on a computer with an NVIDIA GeForce RTX 3090 GPU. Moreover, the standard cross-entropy was used to minimize the training error:(9)$$Loss_{c}=-\frac{1}{N}\overset{N}{\underset{j=1}{\sum }}y^{i}lnP\left(\;y^{i}=1|x^{j}\right)+\left(1-y^{i}\right)lnP\left(\;y^{i}=0|x^{j}\right)$$where *N* represents the number of training samples, $x^{j}$ refers to the *j*th input sequence, and $y^{j}\;$ refers to the label of the *j*th input sequence. We adopted L2 regularization to relieve the overfitting. Therefore, the objective function of TransPhos is defined as:(10)$$min_{W}Loss_{c}+\lambda \sum^{\;}{\left(|\left|W\right|{|}_{2}\right)}^{2}$$where *W* is the L2 norm of the weight matrix and λ is the regularization coefficient. Finally, we adopted the Adam optimizer and the learning rate was set to 0.0002 and the decay was set to 0.00001.

TransPhos can be applied to general phosphorylation site prediction. We explored different hyperparameters and tried to simplify the model design so that it could learn more information between amino acid sequences compared to the reference model. Since many protomer structural parameters easily caused model overfitting when trained on a small dataset, our model performed poorly in kinase phosphorylation site prediction tasks with small amounts of data, so the application of our model to kinase phosphorylation site prediction is not recommended.

### 4.5. Performance Evaluation

The evaluation metrics of protein p-sites can be classified into five methods using different attributes: specificity (SP), sensitivity (SN), accuracy (ACC), the area under the ROC curve (AUC), and the Matthews coefficients of correlation (MCC). These metrics are evaluated with a confusion matrix that compares the actual target values with those predicted by a model. The number of rows and columns in this matrix depends on the number of classes. From the confusion matrix, we identified four values: true positive (TP) indicates the number of positive samples that were correctly classified by the model. False positive (FP) indicates the number of negative samples incorrectly classified by the model. True negative (TN) indicates the number of negative samples correctly classified by the model. False negative (FN) indicates the number of positive samples incorrectly classified by the model.

The ACC metric is defined in Equation (11) as the ratio of the number of all correctly predicted samples to the total number of samples:(11)$$\mathrm{Accuracy}\;=\frac{TP+TN}{TP+TN+FP+FN}$$

The SN or recall is the proportion of true positive prediction to all positive cases: (12)
(12)$$\mathrm{SN}=\mathrm{Recall}\;=\frac{TP}{TP+FN}$$

The SP is defined in Equation (13). It calculates the proportion of samples that were predicted to be true to all negative samples:(13)$$\mathrm{Specificity}\;=\frac{TN}{TN+FP}$$

The precision metric is defined in Equation (14). It calculates the proportion of true positive samples to all cases that were predicted as positive:(14)$$\mathrm{Precision}\;=\frac{TP}{TP+FP}$$

The F1-score is defined in Equation (15). This metric facilitates the process. It can be used to compare the performance of methods with a single number:(15)$$F1=\frac{2\times Precision\times Recall}{Precision+Recall}$$

Two SN and SP measures were used to plot the ROC curve. AUC can evaluate the predictive performance of the model. Furthermore, we also calculated the Mathews’ correlation coefficient between the predicted and true values. A higher correlation represents a better prediction result:(16)$$\mathrm{MCC}\;=\frac{TP\times TN-FP\times FN}{\sqrt{\left(TP+FN\right)\left(TP+FP\right)\left(TN+FN\right)\left(TN+FP\right)}}$$

## 5. Conclusions

A general phosphorylation site prediction approach, TransPhos, was constructed using a transformer encoder architecture and DC-CNN blocks. TransPhos achieved AUC values of 0.8579, 0.8335, and 0.6953 for S, T, and Y phosphorylation sites, respectively, on P.ELM with a 10-fold cross-validation. The model was tested on an independent test dataset, and the AUC values were 0.7867, 0.6719, and 0.6009 for S, T, and Y sites, respectively. Besides AUC values, the predictive performance of our method was found to be significantly better than other deep learning models and existing methods. The results of the significance test also prove that our prediction results were significantly different from other models. The experimental results on the independent dataset showed that our model has a better overall performance in the general phosphorylation site prediction task, especially in the prediction of the S/T sites, which is significantly better than other existing tools and the conventional deep learning model.

## Figures and Tables

**Figure 1 ijms-23-04263-f001:**
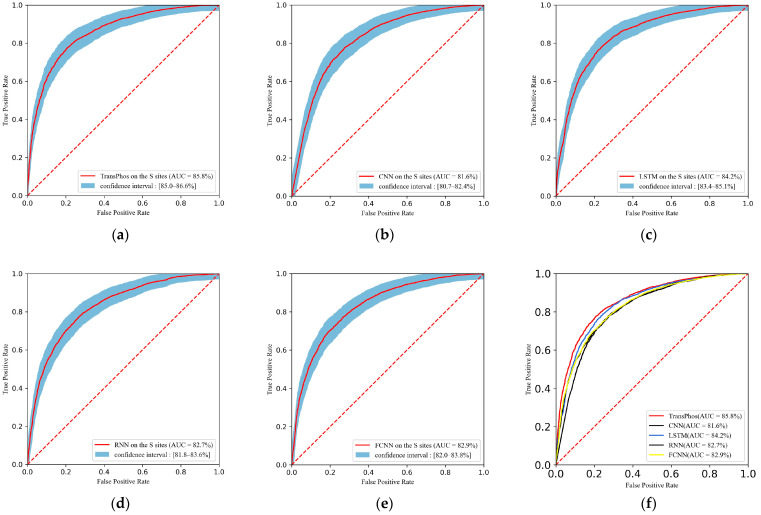
ROC curves containing 95% confidence intervals for different deep learning models on the S sites of the training dataset P.ELM, 10-fold cross validation was used. Area Under Curve (AUC) is defined as the area under the ROC curve to measure the performance of the model. (**a**) ROC curve of the TransPhos model; (**b**) ROC curve of the Convolution neural network (CNN) model. (**c**) ROC curve of the Long and short term memory network (LSTM) model. (**d**) ROC curve of the Recurrent Neural Networks (RNN) model. (**e**) ROC curve of the Fully connected neural networks (FCNN) model. (**f**) Performance comparison on the S sites of the P.ELM dataset.

**Figure 2 ijms-23-04263-f002:**
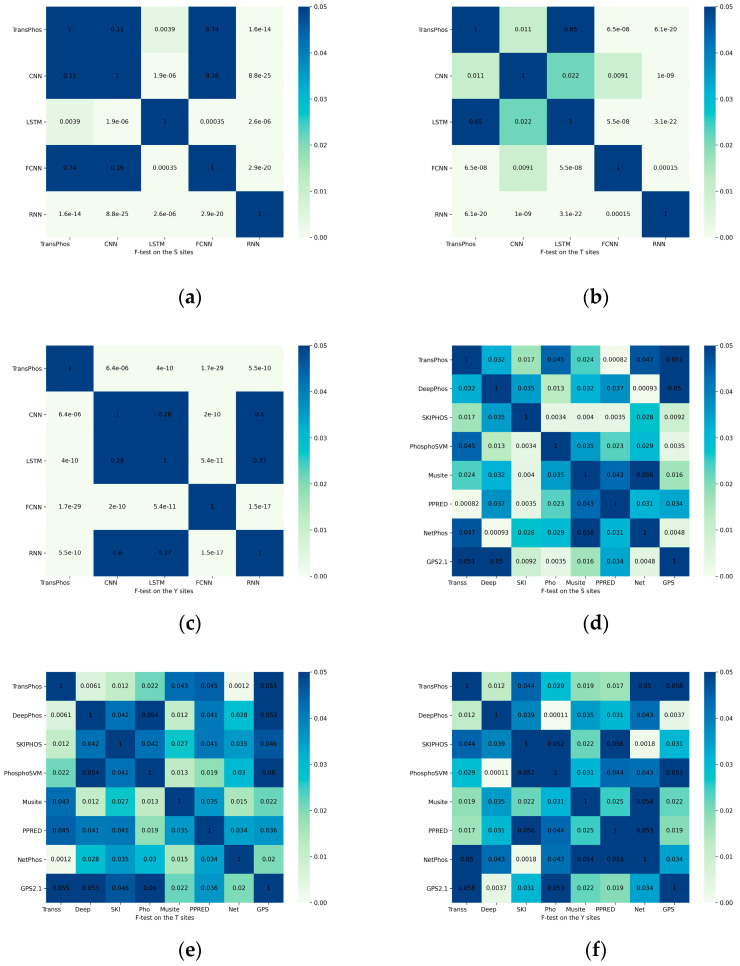
Heat map of the significance F-test, the value of each square in the graph is the *p*-value of the statistical test, and it is generally accepted that a *p*-value less than 0.05 means that the 2 statistics are significantly different. Here we use scientific notation, for example 1.6e-14 means $1.6\times 10^{-14}$. All statistical tests were performed on the predicted results of the test dataset PPA. In the horizontal coordinates, the names of some models are abbreviated to show them in full. (**a**) Significance F-test of the prediction results between the deep learning models for the S sites. (**b**) Significance F-test of the prediction results between the deep learning models for the T sites. (**c**) Significance F-test of the prediction results between the deep learning models for the Y sites. (**d**) Significance F-test of the prediction results for the S sites between other prediction models. (**e**) Significance F-test of the prediction results for the T sites between other prediction models. (**f**) Significance F-test of the prediction results for the Y sites between other prediction models.

**Figure 3 ijms-23-04263-f003:**
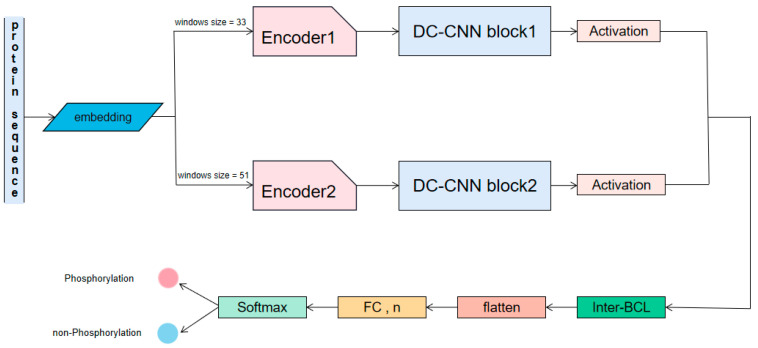
The overall framework of TransPhos. The original sequence is converted into a set of feature vectors with different window sizes through an embedding layer. Here, we set 2 different window sizes: 51 and 33. The sequence features are further represented by the encoder, and then the high-dimensional features are extracted through several densely connected convolutional neural networks (DC-CNN) blocks. After the activation function, the representations obtained by several DC-CNN blocks are concatenated by intra-block connectivity layer (Inter-BCL) and converted to a one-dimensional tensor by a flatten layer. After a full connection (FC) layer, the phosphorylation prediction is finally generated by the SoftMax function.

**Figure 4 ijms-23-04263-f004:**
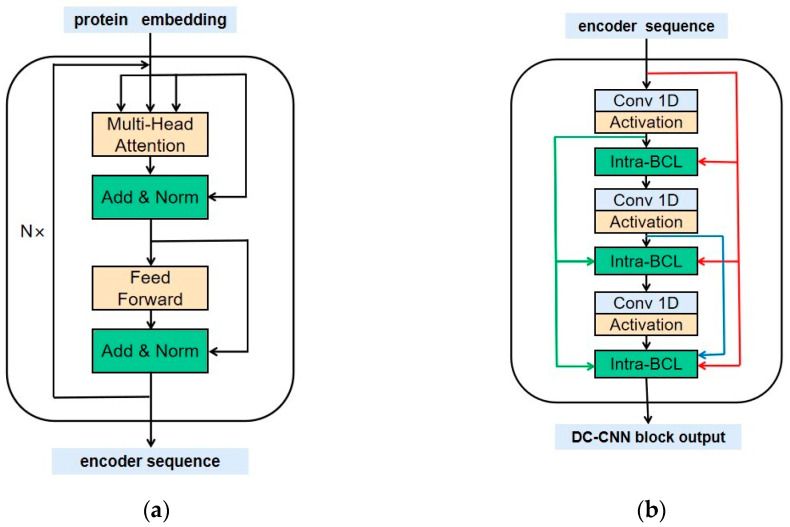
(**a**) The internal construction of an encoder. The encoder is connected by N coding layers with the same structure, where N is set to 4. Each encode layer is composed of two sub-layers. The first sub-layer is a multi-head attention mechanism [25] and here it has four heads. The second sub-layer is a feed-forward neural network. A residual connection [26] is used to connect the two sub-layers, followed by a layer normalization [48]. (**b**)The internal structure of the densely connected convolutional neural network block is the so-called DC-CNN block. Conv1D means one-dimensional convolution. The output sequence of the encoder is converted into a group of sequence feature maps by the densely connected convolution operation. Intra-BCLs between two convolutional layers in each DC-CNN block are used to connect the previous and current feature maps [27].

**Figure 5 ijms-23-04263-f005:**
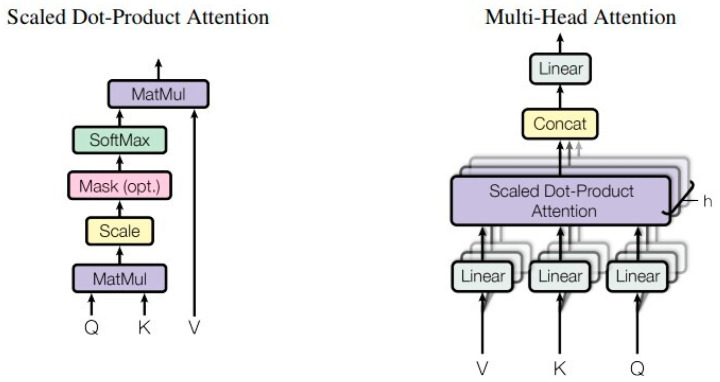
The details of self-attention and multi-head attention (The figure was adapted with permission from Ref. [25]).

**Table 1 ijms-23-04263-t001:** Performance comparison of various deep learning models on the training dataset P.ELM, ten-fold cross validation was used.

Methods	Residue = S						
	AUC (%)	Acc (%)	Sn (%)	Sp (%)	Pre (%)	F1 (%)	MCC
TransPhos	**85.79**	**78.18**	**80.56**	75.80	**76.83**	**78.65**	**0.564**
CNN	81.56	74.96	77.12	72.80	73.85	75.45	0.500
LSTM	84.20	76.99	79.61	74.37	75.57	77.54	0.541
RNN	82.66	75.18	75.39	74.97	75.00	75.20	0.504
FCNN	82.89	75.05	72.93	**77.16**	76.08	74.47	0.501
**Methods**	**Residue = T**						
	**AUC**	**Acc**	**Sn (%)**	**Sp (%)**	**Pre (%)**	**F1 (%)**	**MCC**
TransPhos	**83.35**	75.59	**76.54**	74.70	74.12	75.31	0.512
CNN	81.99	75.50	74.82	76.16	74.82	74.82	0.510
LSTM	83.91	**76.87**	76.09	**77.62**	**76.30**	**76.19**	**0.537**
RNN	79.89	71.72	76.18	67.50	68.93	72.38	0.438
FCNN	80.00	73.48	73.46	73.50	72.41	72.93	0.469
**Methods**	**Residue = Y**						
	**AUC**	**Acc**	**Sn (%)**	**Sp (%)**	**Pre (%)**	**F1 (%)**	**MCC**
TransPhos	69.53	63.62	61.99	65.11	61.99	**69.06**	**0.449**
CNN	67.40	**64.43**	56.17	**72.00**	**64.80**	60.18	0.286
LSTM	68.71	63.73	66.10	61.56	61.21	63.56	0.276
RNN	67.84	62.22	**75.79**	49.78	58.07	65.76	0.264
FCNN	**69.55**	64.31	61.02	67.33	63.16	62.07	0.284

Accuracy (Acc), Sensitivity (Sn), Specificity (Sp), Precision (Pre), F1 Score (F1) and Matthews correlation coefficient (MCC) were calculated to measure the performance of models. Data in bold indicates that the model performs best for that evaluation metric.

**Table 2 ijms-23-04263-t002:** Performance comparison of various deep learning models on the training dataset P.ELM, 10-fold cross validation was used.

Methods	Residue = S						
	AUC (%)	Acc (%)	Sn (%)	Sp (%)	Pre (%)	F1 (%)	MCC
TransPhos	**78.67**	**71.53**	**67.16**	75.89	**73.59**	**70.23**	**0.432**
CNN	74.34	68.40	61.14	75.65	71.52	65.93	0.372
LSTM	77.04	70.48	65.01	75.95	72.99	68.77	0.412
RNN	75.53	68.84	61.44	76.24	72.11	66.35	0.381
FCNN	75.30	69.14	60.68	**77.61**	73.04	66.29	0.388
**Methods**	**Residue = T**						
	**AUC**	**Acc**	**Sn (%)**	**Sp (%)**	**Pre (%)**	**F1 (%)**	**MCC**
TransPhos	**67.19**	**61.77**	47.32	76.22	66.56	55.32	**0.246**
CNN	64.44	59.19	42.03	76.34	63.98	50.74	0.196
LSTM	66.59	60.64	41.85	**79.43**	**67.05**	51.54	0.230
RNN	66.03	61.21	**48.57**	73.84	65.00	**55.60**	0.232
FCNN	63.94	59.63	45.30	73.96	63.50	52.88	0.201
**Methods**	**Residue = Y**						
	**AUC**	**Acc**	**Sn (%)**	**Sp (%)**	**Pre (%)**	**F1 (%)**	**MCC**
TransPhos	60.09	55.41	38.52	72.30	58.17	46.35	0.115
CNN	59.11	54.59	34.81	**74.37**	57.60	43.40	0.100
LSTM	59.49	55.56	40.74	70.37	57.89	47.83	0.116
RNN	**61.71**	**59.48**	**58.96**	60.00	**59.58**	**59.27**	**0.190**
FCNN	59.30	56.44	43.26	69.63	58.75	49.83	0.134

Accuracy (Acc), Sensitivity (Sn), Specificity (Sp), Precision (Pre), F1 Score (F1) and Matthews correlation coefficient (MCC) were calculated to measure the performance of models. Data in bold indicates that the model performs best for that evaluation metric.

**Table 3 ijms-23-04263-t003:** Performance comparison with other predictors on training and independent datasets.

Residue	Methods	10-Fold Cross-Validation Test (P.ELM)				Independent Dataset Test (PPA)			
		Sn	Sp	MCC	AUC	Sn	Sp	MCC	AUC
**S**	GPS 2.1	33.07	93.29	0.201	0.741	22.20	95.26	0.135	0.670
	NetPhos	34.14	86.73	0.123	0.702	28.55	87.23	0.081	0.643
	PPRED	32.27	91.64	0.169	0.751	21.32	94.00	0.107	0.676
	Musite	41.37	93.66	0.249	0.807	28.60	95.21	0.182	0.726
	PhosphoSVM	44.43	**94.04**	0.298	0.841	34.01	**95.90**	0.237	0.776
	SKIPHOS	78.50	74.90	0.521	0.845	46.20	68.60	0.265	0.691
	DeepPhos	**81.81**	75.30	**0.572**	**0.859**	66.43	75.89	0.425	0.775
	**TransPhos**	80.56	75.80	0.564	0.858	**67.16**	75.89	**0.432**	**0.787**
**T**	GPS 2.1	38.10	92.30	0.201	0.695	13.48	94.51	0.067	0.572
	NetPhos	34.32	83.65	0.090	0.655	27.02	80.66	0.038	0.554
	PPRED	30.31	90.99	0.134	0.726	26.43	83.51	0.052	0.578
	Musite	33.84	94.76	0.221	0.785	15.56	**95.36**	0.098	0.622
	PhosphoSVM	37.31	**94.99**	0.251	0.818	21.79	93.41	0.115	0.665
	SKIPHOS	74.40	78.80	**0.547**	**0.844**	**65.80**	58.60	0.197	0.643
	DeepPhos	**77.63**	73.58	0.512	0.826	46.02	76.04	0.231	**0.674**
	**TransPhos**	76.54	74.70	0.512	0.834	47.32	76.22	**0.246**	0.672
**Y**	GPS 2.1	34.49	78.86	0.083	0.611	47.93	60.83	0.043	0.552
	NetPhos	34.66	84.45	0.132	0.653	**63.91**	46.10	0.048	0.554
	PPRED	43.04	82.65	0.169	0.702	42.01	65.08	0.064	0.539
	Musite	38.42	86.74	0.182	0.720	28.85	81.71	0.064	0.587
	PhosphoSVM	41.92	**87.34**	0.209	**0.738**	28.55	**84.39**	0.084	0.595
	SKIPHOS	**71.10**	69.10	**0.396**	0.700	65.80	58.60	**0.197**	**0.634**
	DeepPhos	69.01	64.22	0.332	0.714	49.93	66.37	0.165	0.621
	**TransPhos**	61.99	65.11	0.271	0.695	38.52	72.30	0.115	0.601

The left half is the result of 10-fold cross-validation on the training dataset, and the right half is the result on the independent test set. Sensitivity (Sn), Specificity (Sp), Matthews correlation coefficient (MCC) and Area under curve (AUC) were calculated to measure the performance of models. Data in bold indicates that the model performs best for that evaluation metric.

**Table 4 ijms-23-04263-t004:** The numbers of protein sequences and known phosphorylation sites used in this study in the P.ELM and PPA dataset.

Dataset	Residue	# of Sequences	# of Sites
P.ELM	S	6635	20,964
	T	3227	5685
	Y	1392	2163
PPA	S	3037	5437
	T	1359	1686
	Y	617	676

PPA version 3.0 and Phospho. ELM (P.ELM) version 9.0 were used in this study. The amino acid residues are serine (S) threonine (T) and tyrosine (Y).

## Data Availability

Not applicable.

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
