# Peer review of "TransPhos: A Deep-Learning Model for General Phosphorylation Site Prediction Based on Transformer-Encoder Architecture"

_ijms, 2022, doi:10.3390/ijms23084263_

Round 1

Reviewer 1 Report

In this study, the authors proposed a deep learning model to improve the predictive performance of protein phosphorylation sites. Although the model showed promising performance on different sites, some major concerns should be addressed as follows:

1. For such kind of study, the authors must develop and release a web server for users to test their own sequences.

2. Was the comparison among different predictors (Table 3) performed on the same dataset? It needs to be confirmed to have a fair comparison.

3. "Discussion" section should be improved. This section should be used to discuss their findings and compare them to previous studies, but here the authors did not refer to any references.

4. From the results, the differences between TransPhos and other models were not really significant, so how to know that TransPos performed better than the others? In this case, the authors are suggested to perform some statistical tests to show the significant differences.

5. Uncertainties of models should be reported.

6. Confident interval should be shown in ROC curves.

7. Measurement metrics (i.e., sens, spec, ACC, ...) are well-known and have been used in previous bioinformatics studies such as PMID: 31987913, PMID: 31380767. Therefore, the authors are suggested to refer to more works in this description to attract a broader readership.

8. Why did the authors not combine Fig. 2 and Fig. 1?

9. Quality of figures should be improved.

Reviewer 2 Report

The manuscript titled “TransPhos: a deep learning model for general phosphorylation site prediction based on transformer-encoder architecture” showed the novel transformer encoder and deep CNN for phosphorylation sites in independent datasets. The manuscript is perfectly planned and conducted, and hence should be accepted for publication after author’s update the GitHub link and properly annotate the code.

Author Response

Dear Reviewer
        We would like to thank you and the anonymous reviewers for your valuable comments and constructive suggestions in this paper. Our github link has been placed at the end of the abstract at https://github.com/flyeagle0/TransPhos
We have updated all the code comments in detail and explained how to use our tools in the readme file. In addition, we have developed a simple website so that bioinformatics researchers can quickly predict their data without having to configure the environment themselves, but currently users are still unable to train their models through the website, and we will consider adding a training feature to it later. Finally, thank you again for your appreciation of our work.

Round 2

Reviewer 1 Report

Thanks for addressing my previous comments. However, some comments were not addressed well as follows:

1. About web server, I have accessed and seen that it did not work, thus the authors should at least make it work normally before submission. A webserver is mandatory for such kind of study.

2. Still, the comparison among different predictors was not done on the same dataset. Thus it is not a fair comparison. In this way, the authors should reproduce the previous model and apply it to the current data.

3. Uncertainties of models should be reported.

4. Confident interval should be shown in ROC curves. The authors mentioned that CI is not shown in such results, however, this point is unacceptable. We may easy to find that many papers use 95% CI in reporting results in this field such as https://doi.org/10.1093/bib/bbab396, https://doi.org/10.1093/bib/bbab095, https://doi.org/10.1093/bib/bbu054, etc.

5. Some previous comments were not addressed i.e., #4,#7.

Round 3

Reviewer 1 Report

My previous comments have been addressed.